# The Role of Adsorption in the Photocatalytic Decomposition of Dyes on APTES-Modified TiO₂ Nanomaterials

**Ewelina Kusiak-Nejman \*** , **Agnieszka Sienkiewicz, Agnieszka Wanag** , **Paulina Rokicka-Konieczna and Antoni W. Morawski**

Department of Inorganic Chemical Technology and Environment Engineering, Faculty of Chemical Technology and Engineering, West Pomeranian University of Technology in Szczecin, Pułaskiego 10, 70-322 Szczecin, Poland; Agnieszka.Sienkiewicz@zut.edu.pl (A.S.); awanag@zut.edu.pl (A.W.); prokicka@zut.edu.pl (P.R.-K.); amor@zut.edu.pl (A.W.M.)

\* Correspondence: ekusiak@zut.edu.pl; Tel.: +48-91-449-42-44

**Abstract:** This work investigated for the first time the role of adsorption in the photocatalytic degradation of methylene blue and Orange II dyes in the presence of 3-aminopropyltriethoxysilane (APTES)-modified TiO₂ nanomaterials. It has been demonstrated that the decrease in adsorption has a detrimental effect on photocatalytic activity. APTES/TiO₂ photocatalysts were successfully prepared by solvothermal modification of TiO₂ in a pressure autoclave, followed by heat treatment in an inert gas atmosphere at the temperature range from 300 °C to 900 °C. It was observed that functionalization of TiO₂ via APTES effectively suppressed the anatase-to-rutile phase transformation, as well as the growth of crystallites size during calcination, and reduction of specific surface area (APTES modification inhibits sintering of crystallites). The noted alterations in the adsorption properties, observed after the calcination, were generally related to changes in the surface characteristics, mainly surface charges expressed by the zeta potential. Positively charged surface enhances adsorption of anionic dye (Orange II), while negatively charged surface was better for adsorption of cationic dye (methylene blue). The adsorption process substantially affects the efficiency of the photocatalytic oxidation of both dyes. The methylene blue decomposition proceeded according to the pseudo-first and pseudo-second-order kinetic models, while the degradation of Orange II followed the zero, pseudo-first, and pseudo-second order kinetic models.

**Keywords:** titanium dioxide; 3-aminopropyltriethoxysilane; calcination; dyes photodegradation; adsorption capacity





## 1. Introduction

In addition to consuming large amounts of water, the textile industry also uses a wide range of synthetic dyes, as a result of which it contributes to the discharge of a large quantity of wastewater. In the states of Gujarat and Maharashtra, India, the annual production of synthetic dyes for textiles and other industries is about 130,000 tons, and due to the high toxicity, solubility, mutagenicity and lack of biodegradability, these colourants are one of the most problematic and hazardous water pollutants [1]. Therefore, many researchers are trying to develop the best method to remove different types of dye from water.

Currently, the elimination of this organic contaminant is carried out using various techniques such as flocculation and coagulation [2,3], biological oxidation [4,5], adsorption [6,7], and membrane filtration [5,8]. Recent studies also focus on using advanced oxidation processes (AOPs) to decompose synthetic dyes present in water. Among AOPs, for example, H₂O₂/ultraviolet (UV) processes [9] or Fenton and photo-Fenton catalytic reactions [10,11], heterogeneous photocatalysis using titanium dioxide as a photocatalyst appear to be one of the most influential technologies [12,13]. The main advantages are that photocatalysis could be carried out under ambient conditions, does not involve mass transfer, and can also lead to total transformation of organic carbon to CO₂ [14]. Additionally,

$TiO_2$ is widely used due to its low cost, chemical and biological stability, low toxicity, and high photoactivity [15].

It is commonly known that photocatalytic reactions occur primarily on the surface of the semiconductor. Therefore, the adsorption of contaminants on the surface of the material plays a crucial role in affecting the photodegradation efficiency [16]. Many experiments measured the adsorption and the kinetics of photodecomposition of various dyes on $TiO_2$, for instance, Reactive Black [17], Rhodamine B [18,19], Orange II [20], methylene blue [17,21], Direct Green 99 [22] and thionine [19]. In all presented research, organic compounds are adsorbed on the surface of the semiconductor, until the adsorption-desorption equilibrium between the dye molecules and the $TiO_2$ surface is reached. After that, the lamps are turned on, and the photooxidation is carried out [22]. Moreover, Zhu et al. [23] noted that the high adsorption capacity of polythiophene/$TiO_2$ photocatalyst could potentially increase the rate of photocatalytic decomposition of methyl orange. Wang et al. [24] also reported that the enhancement of the degradation kinetics of dyes after fluorination of $TiO_2$ was due to increased adsorption and decreased flat band potential.

Additionally, $TiO_2$ surface properties are also crucial for the effectiveness of photooxidation. Therefore, photocatalytic studies' significant field concentrates on modifying $TiO_2$ to improve its physicochemical properties and photocatalytic activity [21]. In recent years, organosilanes such as 3-aminopropyltriethoxysilane (APTES), as surface modifiers, have received increasing research interest, since most applications of modified nanomaterials require them to be chemically and thermally stable, well dispersed in the media and characterized by a large specific surface area [25,26]. For example, APTES-modified $TiO_2$ nanoparticles have been used to induce the self-cleaning capability of ceramic tiles [27] or-enhance the decomposition degree of Reactive Brilliant Red X-3B [28]. Moreover, the adsorption properties of organic dyes (Acid Orange 7 and Reactive Blue 19) on the APTES-functionalized $TiO_2$ pigment surface was also investigated [29]. In this case, the modified rutile titanium oxide was used as an adsorbent, and due to the weak photoactivity of this polymorphous form of $TiO_2$ no photocatalytic degradation of tested dyes was performed.

In the present study, the APTES-modified $TiO_2$ nanomaterials were obtained by a solvothermal process, and then calcination in an inert gas atmosphere. This combined method has been proposed for the first time by our research group [30]. Other preparation methods described in the literature, where APTES-modified $TiO_2$ were utilized in the removal process of dyes by adsorption or photooxidation, involve the sol-gel method [28] and simple mixing of $TiO_2$ slurry with aminosilane coupling agents [29]. Here, two commercial textile dyes: cationic methylene blue and anionic Orange II, were removed from aqueous solution. To the best of our knowledge, this is the first paper presenting the results of research on the role of adsorption in the photocatalytic degradation of methylene blue and Orange II in the presence of APTES-modified $TiO_2$ photocatalysts. The novelty of the presented studies was also to establish the impact of modifying the efficiency and potential reuse of the prepared photocatalysts during subsequent dye decomposition cycles.

## 2. Results and Discussion

### 2.1. Characterization of Materials

X-ray diffraction (XRD) patterns of APTES/$TiO_2$ nanomaterials are presented in Figure 1. The structural characteristics of the starting-$TiO_2$ and calcined reference samples were discussed in detail in our previously published work [30]. Except for the sample calcined at 900 °C, all tested nanomaterials showed reflections characteristic of the anatase phase located at 25.3, 37.8, 48.1, 53.9, 55.1, 62.7, 68.9, 70.3 and 75.1°, which correspond to (101), (004), (200), (105), (211), (204), (116), (220), (215) indexed by JCPDS 01-070-7348, and certain reflections characteristic of the rutile phase located in 27.4, 36.0 and 41.2°, corresponding to (110), (101) and (111) indexed by JCPDS 01-076-0318. While the $TiO_2$-Ar-900°C sample demonstrated only reflections characteristic for the rutile phase located at 27.4, 36.0, 39.1, 41.2, 44.0, 54.3, 56.6, 62.7, 64.0, 69.0 and 69.7°, which correspond to (110), (101), (200), (111), (210), (211), (220), (002), (310), (301) and (112), respectively. The

presence of rutile in the starting-$TiO_2$ resulted from the addition of rutile nuclei during the production process of $TiO_2$ pulp via sulphate technology. For starting-$TiO_2$, the anatase-to-rutile phase transformation typically occurs between 600 °C and 700 °C [31,32]. In our case, the phase transformation started above 500 °C, and starting $TiO_2$ was fully transformed into rutile at 900 °C [30]. According to the XRD phase composition and the crystallites size data shown in Table 1, the amount of anatase in APTES-modified $TiO_2$ samples was constant, and its content was about 96%. Moreover, it is important to note that after calcination at 900 °C, the $TiO_2$-4h–120°C–500mM-Ar–900°C sample still contained 95% of anatase phase. Silicon present in the APTES contributed to the effective suppression of anatase-to-rutile phase transformation during thermal modification [33–35]. For all tested photocatalysts, the crystallites size enhanced with the increase of the modification temperature. For starting-$TiO_2$ and calcined reference samples, the crystallites size of anatase was in the range of 14–52 nm [30], and 14–30 nm for APTES/$TiO_2$ nanomaterials. Whereas, the crystallites size of rutile for starting-$TiO_2$ and reference samples was from 21 nm to >100 nm, and from 46 nm to 79 nm for APTES/$TiO_2$ nanomaterials. Nevertheless, comparing the crystallites size of photocatalysts calcined at the same temperature with and without APTES, the crystallites size of both polymorphous forms of $TiO_2$ was significantly smaller for APTES-modified samples than for the reference materials. For example, the anatase crystallite size for $TiO_2$-4h–120°C–500mM-Ar–700°C sample equalled only 17 nm, while for $TiO_2$-Ar–700°C it was 52 nm [30]. According to Dalod et al. [33], the use of such stabilizers as APTES can lead to the formation of an amorphous $SiO_2$ layer on the surface, derived from the organosilane coupling agent, and leading to $TiO_2$-$SiO_2$ core-shell nanostructures, so that, during the thermal modification both the growth of the crystallites size and phase transformation can be successfully inhibited. Furthermore, the results presented by Lu et al. [35] and Xu et al. [36] also showed that silicon could effectively suppress the growth of titania grains over the calcination process.

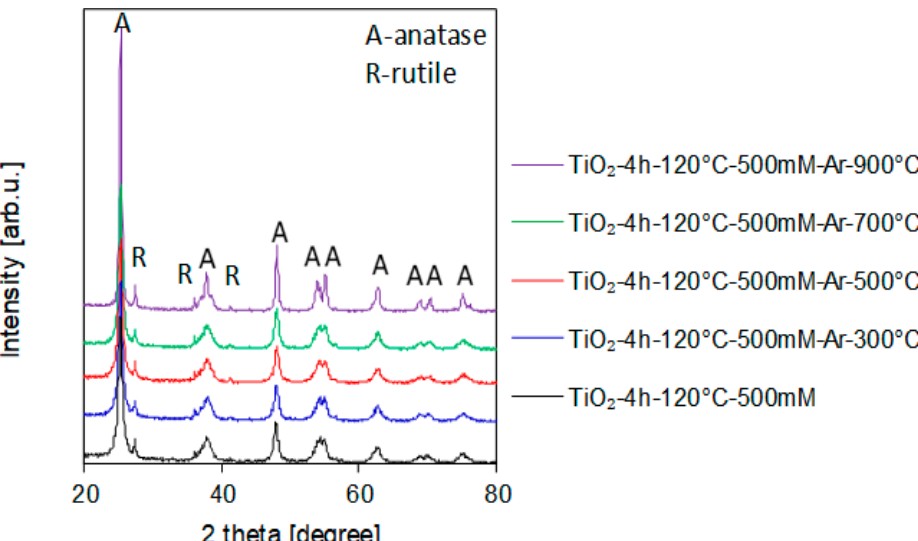

**Figure 1.** X-ray diffraction (XRD) patterns of 3-aminopropyltriethoxysilane (APTES)-modified $TiO_2$ before and after the calcination process.

**Table 1.** Structural parameters of starting $TiO_2$ and APTES-modified photocatalysts prior and after the calcination process.

| Sample Name | Anatase in Crystallite Phase (%) | Anatase Crystallite Size (nm) | Rutile in Crystallite Phase (%) | Rutile Crystallite Size (nm) | $S_{BET}$ (m²/g) | $V_{total}$ (cm³/g) | $V_{micro}$ (cm³/g) | $V_{meso}$ (cm³/g) |
|---|---|---|---|---|---|---|---|---|
| starting $TiO_2$ | 95 | 14 | 5 | 21 | 207 | 0.370 | 0.070 | 0.300 |
| $TiO_2$–4h–120°C–500mM | 96 | 14 | 4 | 46 | 140 | 0.272 | 0.053 | 0.219 |
| $TiO_2$–4h–120°C–500mM-Ar-300°C | 95 | 14 | 5 | 36 | 158 | 0.242 | 0.058 | 0.184 |
| $TiO_2$–4h–120°C–500mM-Ar-500 °C | 96 | 15 | 4 | 51 | 170 | 0.272 | 0.062 | 0.210 |
| $TiO_2$–4h–120°C–500mM–Ar–700°C | 96 | 17 | 4 | 45 | 140 | 0.267 | 0.053 | 0.214 |
| $TiO_2$–4h–120°C–500mM-Ar-900°C | 95 | 30 | 5 | 79 | 50 | 0.150 | 0.019 | 0.131 |

In Figure 2 the adsorption-desorption isotherms of the obtained photocatalysts were shown. Most of them presented a type IV isotherm in IUPAC classification, characteristic for mesoporous materials [37]. Other nanomaterials presented the H3 type of the hysteresis loops. The H3 loops have characteristic desorption shoulders, lower closure points, and no plateau in the high $p/p_0$ value [38,39]. The pore size distribution was presented in Figure S1 (in the Supplementary Materials section).

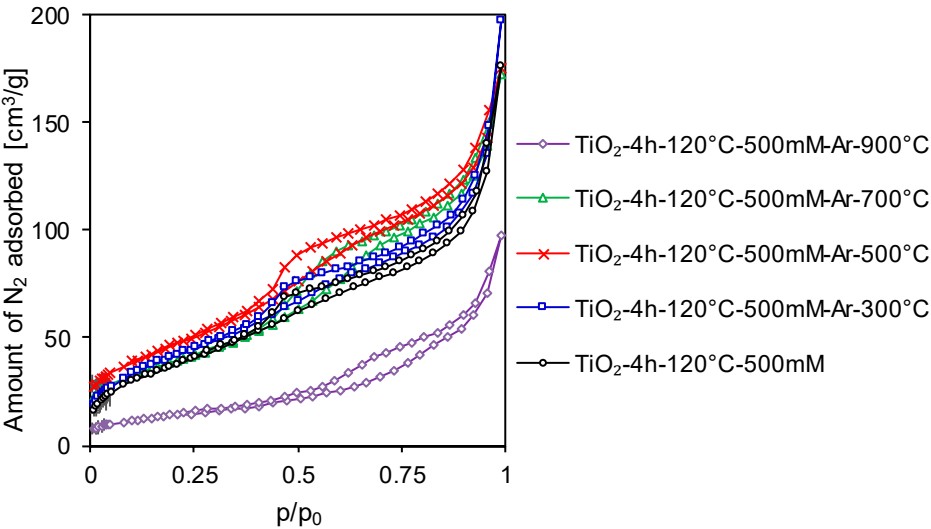

**Figure 2.** Adsorption/desorption isotherms of APTES-modified $TiO_2$ prior and after calcination process.

Furthermore, according to the data exhibited in Table 1, most of the tested semiconductors were mesoporous materials with a small number of micropores (except for the $TiO_2$-Ar–900°C sample that was a non-porous material, as presented in our previous paper [30]). In starting $TiO_2$, after modification with 500 mM of APTES, a significant decrease of the specific surface area and total pore volume was noted. Compared to the starting-$TiO_2$, the $S_{BET}$ of $TiO_2$-4h–120°C–500mM reduced by 67 m²/g and $V_{total}$ by 0.098 cm³/g. Dalod et al. [33] observed that $TiO_2$ surface functionalization using different silane coupling agents such as 3-aminopropyltriethoxysilane, 3-(2-aminoethylamino)propyl- dimethoxy-methylsilane and *n*-decyltriethoxysilane decreased the specific surface area, from 195 m²/g for unmodified $TiO_2$ to 178 m²/g, 149 m²/g and 114 m²/g, respectively. Siwińska-Stefańska et al. [40] also noted that modification via different alkoxysilanes compounds such as vinyltrimethoxysilane, n-2-(aminoethyl)-3-aminopropyltrimethoxy- silane and 3-methacryloxypropyltrimethoxysilane, caused a decrease of the $S_{BET}$ and $V_{total}$ of obtained nanomaterials. This was most likely because the modifier particles blocked the active centers on the $TiO_2$-$SiO_2$ surface. Zhuang et al. [41] also found that the specific surface area and the pore volume were smaller for the nanocomposites than for the unmodified $TiO_2$ because APTES also penetrated inside the pores, rather than only dispersing on the external surface.

After calcination, the $S_{BET}$ of APTES-modified $TiO_2$ increased from 140 m$^2$/g for $TiO_2$–4 h–120 °C–500mM through 158 m$^2$/g for the sample calcined at 300 °C to 170 m$^2$/g for heated at 500 °C. Diminishing and eventually disappearing APTES-characteristic peaks with the increase of the modification temperature, observed on diffuse reflectance Fourier-transform infrared spectra (DRIFTS), presented in Figure 3, as well as the observed significant decrease in carbon and nitrogen content (see Table 2) indicated the unblocking of the surface of the nanomaterials. Therefore, the increase in the specific surface area was related to the decomposition of APTES molecules during the calcination process. The increase of $V_{total}$ value from 0.242 cm$^3$/g (for sample calcined at 300 °C) to 0.272 cm$^3$/g after thermal modification at 500 °C, also pointed out that the modifier molecules unblocked both the external surface of $TiO_2$ and the pores. As shown in Table 1, a further increase in the modification temperature above 500 °C caused a decrease in the specific surface area and pore volume of the tested photocatalysts, which resulted in the increase of the anatase and rutile crystallites size and sintering of photocatalysts particles [42].

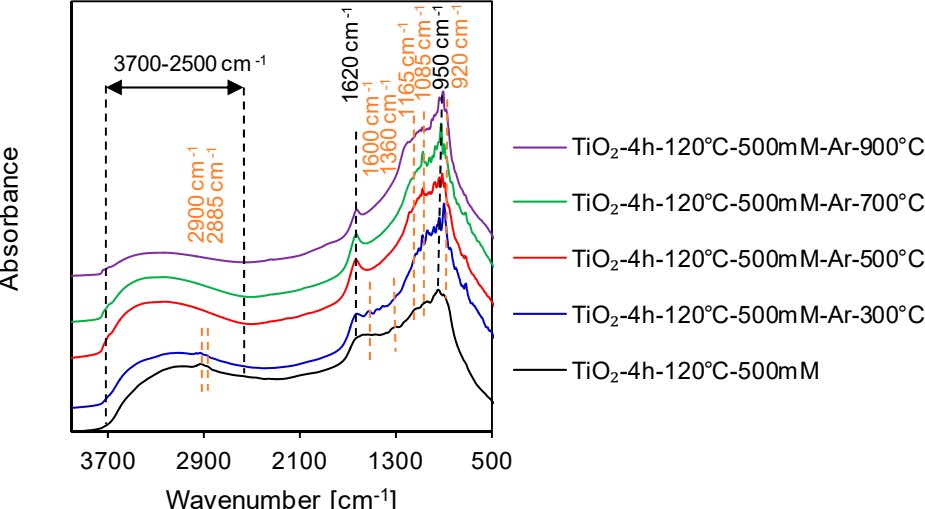

**Figure 3.** Diffuse refletrance Fourier transform infrared spectra (DRIFTS) of and APTES-modified $TiO_2$ before and after calcination process.

The DRIFTS measurements were utilized to determine the surface characteristics of the prepared photocatalysts. All spectra presented in Figure 3 showed a narrow band at 1620 cm$^{-1}$, as well as a wide band from 3700 cm$^{-1}$ to 2500 cm$^{-1}$, associated with the molecular water and stretching vibrations of surface hydroxyl groups [28,43,44]. An increase of the modification temperature of APTES/$TiO_2$ samples resulted in a decrease of the intensity of these bands, due to the changes in the amount of surface $-OH$ groups [45,46]. All materials exhibited the intensive band at around 950 cm$^{-1}$ assigned to the self-absorption of titania [46,47]. Some new characteristic bands of APTES were observed in the spectra, shown in Figure 3, which indicates that the modified photocatalysts were prepared successfully. The bands detected at 2900 cm$^{-1}$ and 2885 cm$^{-1}$, which belongs to the bending and stretching contributions of the alkyl groups [(CH$_n$)] [48–50]. The asymmetric $-NH_3^+$ deformation modes were observed near 1600 cm$^{-1}$ [49,51,52]. Another low-intensity band at 1360 cm$^{-1}$, indicating the existence of C$-$N bonds [51,53]. Bands noted 960–910 cm$^{-1}$ ascribe the stretching vibrations of Ti–O–Si chemical interactions [54,55]. Additionally, the band localized at 920 cm$^{-1}$ indicated the condensation reaction between silanol and $-OH$ groups on the surface of the semiconductor [54]. Above 300 °C, bands characteristic for APTES assigned to alkyl groups, $-NH_3^+$, C$-$N and Ti–O–Si bonds were no longer observed. Furthermore, the band at around 1165 cm$^{-1}$ corresponds to the Si$-$O$-$Si stretching vibrations, created by the condensation reaction between silanol groups [43,51]. The presence of Si$-$O$-$C stretching mode was noted at around 1085 cm$^{-1}$ [51,53].

From the ultraviolet–visible diffuse reflectance spectra (UV–Vis/DRS) of all APTES-modified TiO$_2$, shown in Figure 4, it was noted that TiO$_2$–4h–120°C–500mM exhibited the characteristic absorption in the UV region due to the intrinsic band gap absorption of Ti [56,57]. Moreover, for all APTES/TiO$_2$ photocatalysts calcined in an inert gas conditions, the f reflectance in the visible range decreased as the heating temperature increased, due to the change of colour from white for TiO$_2$-4h–120°C–500mM through greyish for TiO$_2$-4h–120°C–500mM-Ar–500°C, to dark grey for TiO$_2$-4h–120°C–500mM-Ar–900°C material [58,59]. Interestingly, the bang gap energy values determined for all calcined APTES-modified TiO$_2$ samples did not change in comparison to starting titania sample, including nanomaterials calcined at 700 and 900 °C. These results also provide the evidence of the inhibition of phase transition.

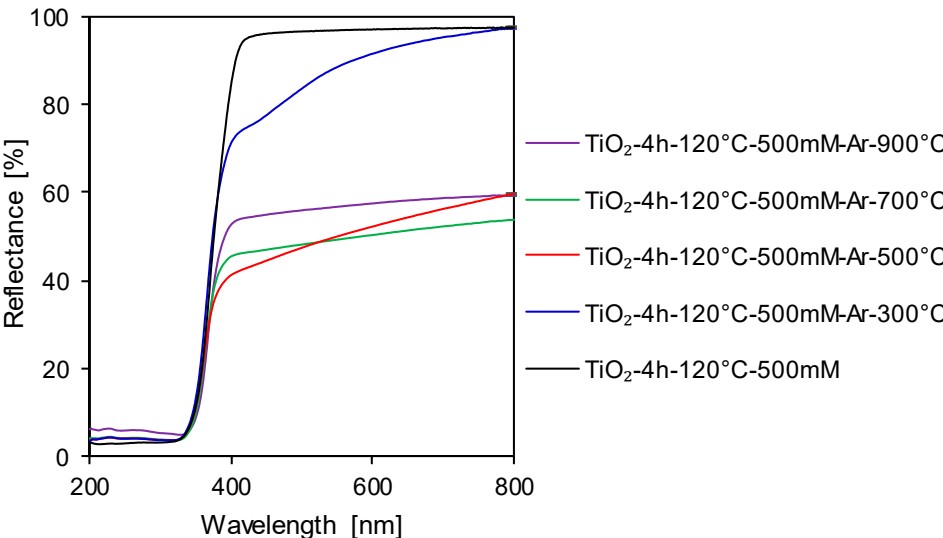

**Figure 4.** Ultraviolet–visible diffuse reflectance spectra (UV–Vis/DRS) of and APTES-modified TiO$_2$ prior and after calcination process.

**Table 2.** The carbon and nitrogen content, and zeta potential values of starting TiO$_2$, reference samples and APTES-modified photocatalysts prior to and after the calcination process.

| Sample Name | Zeta Potential δ (mV) | Carbon Content (wt.%) | Nitrogen Content (wt.%) |
|---|---|---|---|
| starting TiO$_2$ | +11.00 | - | 0.18 |
| TiO$_2$-Ar–300°C | +12.34 | - | - |
| TiO$_2$-Ar–500°C | +12.43 | - | - |
| TiO$_2$-Ar–700°C | +24.77 | - | - |
| TiO$_2$-Ar–900°C | −29.67 | - | - |
| TiO$_2$–4h–120°C–500mM | +4.43 | 3.34 | 1.24 |
| TiO$_2$-4h–120°C-500mM-Ar-300°C | +19.43 | 2.33 | 0.66 |
| TiO$_2$–4h–120°C–500mM-Ar–500°C | −25.63 | 0.50 | 0.14 |
| TiO$_2$–4h–120°C–500mM-Ar–700°C | −25.57 | 0.30 | 0.11 |
| TiO$_2$–4h–120°C–500mM-Ar-900°C | −36.13 | 0.16 | 0.07 |

Following the data presented in Table 2, the occurrence of carbon (3.34 wt.%) and nitrogen (1.24 wt.%) in TiO$_2$-4h–120°C–500mM sample, confirmed the presence of APTES in the material obtained after TiO$_2$ modification. Additionally, it was found that the content of the analyzed elements decreased with the increase of calcination temperature, most probably as a result of nitrogen and carbon decomposition from the solid phase [46,60]. The data obtained from DRIFT spectra (see Figure 3) agreed with the analysis of carbon and nitrogen content, which showed a continued decrease of carbon and nitrogen in

APTES-modified $TiO_2$ samples as the calcination temperature increases. The preparation procedure can explain the presence of nitrogen in the starting-$TiO_2$ (preliminary rinsing with $NH_4OH$), used to eliminate the residual sulfuric acid from the raw slurry of $TiO_2$ obtained by sulphate technology [61].

Based on the data shown in Table 2, the zeta potential values of APTES/$TiO_2$ photocatalysts changed from +4.43 mV for $TiO_2$–4h–120°C–500mM to $-36.13$ mV for $TiO_2$-4h–120°C–500mM-Ar–900°C. Goscianska et al. [62], Ukaji et al. [54] and Talavera-Pech et al. [63] found that the amine groups of APTES tend to gain protons, and thus, $NH_3^+$ species can be formed. The $NH_3^+$ can easily be attached to the $TiO_2$ surface, which results in a positive surface charge of the semiconductor. The FT-IR/DR spectra presented in Figure 3, as well as the noted significant decrease of the nitrogen content (see Table 2), showed that the positively charged amino groups were not observed on the $TiO_2$ surface when the calcination temperature exceeded 300 °C. Above this temperature, mostly silicon groups were found on the $TiO_2$ surface. Wilhelm et al. [64] and Ferreira-Neto et al. [65] noted that silica-modified $TiO_2$ exhibited negative zeta potential values. The zeta potential of APTES/$TiO_2$ nanomaterials prepared via calcination at temperature above 300 °C, changed from positive to negative.

## 2.2. Adsorption Experiments

Before the photocatalytic activity test, studies on determination of the adsorption-desorption equilibrium were performed. The adsorption degree of methylene blue was shown in Figure 5 while the adsorption degree of Orange II was presented in Figure 6A,B. For methylene blue the adsorption-desorption equilibrium was established after 60 min for starting $TiO_2$ and $TiO_2$–4 h–120 °C–500mM samples, whereas after 180 min for calcined reference materials and APTES-modified $TiO_2$ nanomaterials received after the calcination. Different adsorption-desorption equilibrium time is associated with the surface characteristics of semiconductors (see Table 2). For starting $TiO_2$, all calcined reference materials [30], $TiO_2$–4 h–120 °C–500mM and $TiO_2$–4 h–120 °C–500mM-Ar-300°C, the adsorption degree of methylene blue reached c.a. 5%. For other APTES/$TiO_2$ nanomaterials, an increase of the calcination temperature above 300 °C resulted in the notable improvement of adsorption properties. Based on zeta potential values (shown in Table 2), for materials with positively charged surface, the low adsorption of a cationic dye was observed. While, photocatalysts with the negatively charged surface exhibited a good adsorption efficiency, due to the higher potential of contact with the positively charged methylene blue molecules. The highest adsorption degree of methylene blue equalled 30%, was found for the $TiO_2$-4h–120°C–500mM-Ar–900°C sample with the most negative zeta potential value of $-36.13$ mV and the lowest value for specific surface area. Therefore, the change in $TiO_2$ surface charge after calcination from positive to negative was found to be the main factor that enhances the adsorption properties of the APTES-modified $TiO_2$ [66–68]. It can be additionally noted that both photocatalysts calcined at 900 °C exhibited a different dye adsorption degree, although almost the same zeta potential value characterized them (for $TiO_2$-Ar-900°C was only 5%). For this reason, the noted enhancement of the adsorption properties at the highest temperature of modification was mainly associated with the values of specific surface area (50 $m^2/g$ for $TiO_2$-4h–120°C–500mM-Ar-900°C and 3 $m^2/g$ for $TiO_2$-Ar-900°C [30]). It was confirmed that APTES modification successfully prevents the anatase-to-rutile transformation and sintering of the crystallites at a higher temperature of calcination process.

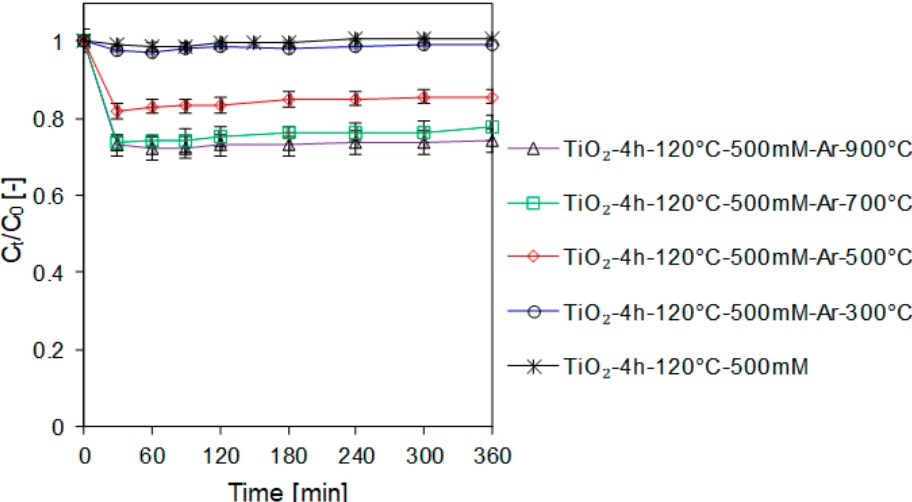

**Figure 5.** Adsorption degree of methylene blue dye on the surface of APTES-modified $TiO_2$ before and after calcination.

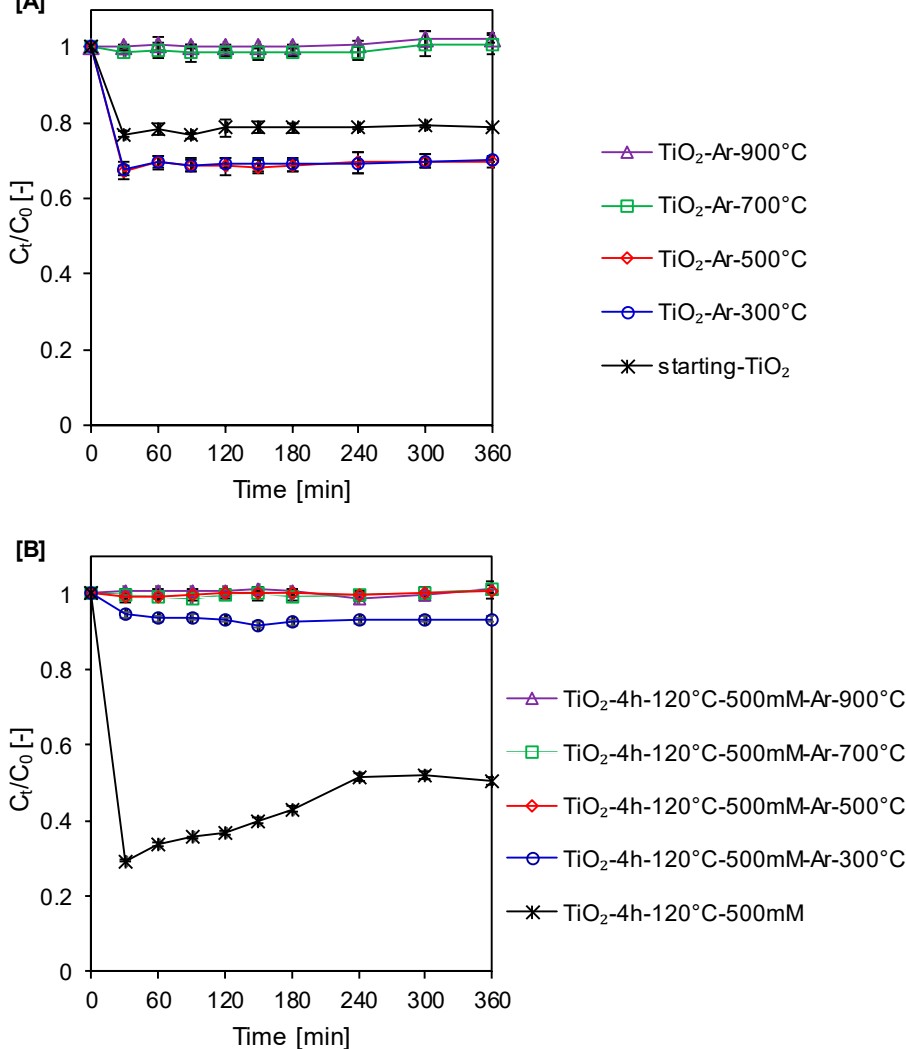

**Figure 6.** Adsorption degree of Orange II dye on the surface of starting $TiO_2$ and calcined reference samples (**A**), and APTES-modified $TiO_2$ before and after calcination process (**B**).

For Orange II, the adsorption-desorption equilibrium was established after 90 min for all calcined reference materials, whereas after 60 min for starting and calcined APTES-modified $TiO_2$ photocatalysts. In case of $TiO_2$-4h–120°C–500mM sample, significant Orange II desorption (approximately 23%) was observed prior to the adsorption-desorption equilibrium established after 240 min. For APTES/$TiO_2$ photocatalysts, the increase of the calcination temperature resulted in a decrease of adsorption properties related to the change of the $TiO_2$ surface charge from positive to negative (see Table 2). The positively charged surface of the semiconductor has a higher potential of contact with the negatively charged dye molecules like Orange II. Therefore, for nanomaterials with a negatively charged surface, the low adsorption efficiency of anionic dye was noted [20,66–69].

Adsorption Capacity

To better understand the adsorption process, additional tests were conducted to determine the adsorption capacity of examined materials. The values of constants and correlation coefficients for Freundlich, Langmuir, Langmuir–Freundlich, and Temkin isotherm models are listed in Table S1 for methylene blue and in Table S2 for Orange II dye.

Based on the correlation coefficients obtained for methylene blue adsorption, it was revealed that the Langmuir and Langmuir–Freundlich isotherm models gave similar and the best simulation of the experimental data with the highest $R^2$ for all APTES-modified $TiO_2$ samples. The Langmuir–Freundlich isotherm model describes most accurately the adsorption properties of starting-$TiO_2$ and all calcined reference photocatalysts, while for $TiO_2$-4h–120°C–500mM the best simulation of the experimental data provided the Temkin isotherm. For photocatalysts that have shown the best fit to the Langmuir–Freundlich isotherm model, it can be assumed that the surface of the semiconductors is heterogeneous [70]. The increase of the adsorption capacity for methylene blue can be explained as the effects of modification, including an increase of the specific surface area (see Table 1), a change in surface charge (see Table 2), and mesoporous structure of APTES/$TiO_2$ nanomaterials [20,71].

Following the calculated correlation coefficients, the experimental equilibrium data for the adsorption of Orange II for starting $TiO_2$ and reference samples calcined at 300–700 °C were most closely fitted to the Freundlich isotherm. $TiO_2$-Ar–900°C and TiO2–4 h–120°C–500mM-Ar-900°C samples were found to be the weakest adsorbents of Orange II dye due to the negative character of the surface (see Table 2), and characterized by negligible adsorption of anionic dye (see Figure 6A,B), which effectively made it very difficult to match the appropriate model of adsorption isotherm. For $TiO_2$–4h–120°C–500mM and APTES-modified $TiO_2$ calcined at 300–700 °C, the best simulation of the experimental data was provided by the Freundlich or Langmuir–Freundlich isotherm model. It can be concluded that the surfaces of APTES-modified semiconductors are heterogeneous [20,71]. The noted increase of the *n* and $q_m$ values was related to the change of surface charge from negative to positive due to $TiO_2$ modification (see Table 2) [20,71,72].

### 2.3. Photocatalytic Activity Test

The photocatalytic activity of the examined nanomaterials was calculated based on the degradation of methylene blue and Orange II dyes under UV light irradiation. Tests performed in the absence of the photocatalyst (see Figures 7A and 8A) revealed that, for both methylene blue and Orange II, the decomposition due to the photolysis process was insignificant compared to the photocatalysis process. After 360 min of exposure under the same conditions as in the photocatalytic activity tests, about 2.5% decomposition of dyes was recorded.

For methylene blue, a significant improvement of the photoactivity from 39% to 95% for starting $TiO_2$ was noted only for the nanomaterial calcined at 700 °C, consisting of 88% of anatase and 12% of rutile phase, and the band gap energy value of 3.07 eV [30]. Ohno et al. [73] observed that $TiO_2$, being an optimal mixture of anatase and rutile phases, exhibited higher photocatalytic activity than $TiO_2$ composed of only one phase. Moreover,

the co-existence of various-forms of $TiO_2$ is essential for determining the $TiO_2$ photoactivity, and the most effective performance during methylene blue degradation was achieved for photocatalysts with slightly reduced $E_g$ (3.04 eV) and a mixture of both phases. Additionally, these samples did not have the largest $S_{BET}$ among all examined materials. After 240 min of exposure to UV light, the methylene blue removal degree determined in the presence of $TiO_2$-4h–120°C–500mM sample compared to starting $TiO_2$, enhanced from 39% to 82% (see Figure 9). After modification of $TiO_2$ via APTES, the increase of the efficiency was essentially associated with the presence of nitrogen and carbon in the materials obtained (see Table 2) [74]. After thermal modification, almost all APTES/$TiO_2$ photocatalysts showed an improved methylene blue decomposition degree in comparison to reference materials heat treated at the same temperature. Only $TiO_2$-4h–120°C–500mM-Ar–700°C sample showed a slight 7% decrease in efficiency compared to $TiO_2$-Ar-700°C. The reported increase in the activity was related to the fact that the modification with APTES, which is a good source of silicon, effectively suppressed the anatase-to-rutile phase transformation, as well as the growth of crystallites size (see Table 1). The growth of the anatase phase crystallinity and the decrease of the bulk defects after thermal modification resulted in an increase of the electron diffusion rate to the surface of the semiconductor by inhibiting electron-hole recombination.

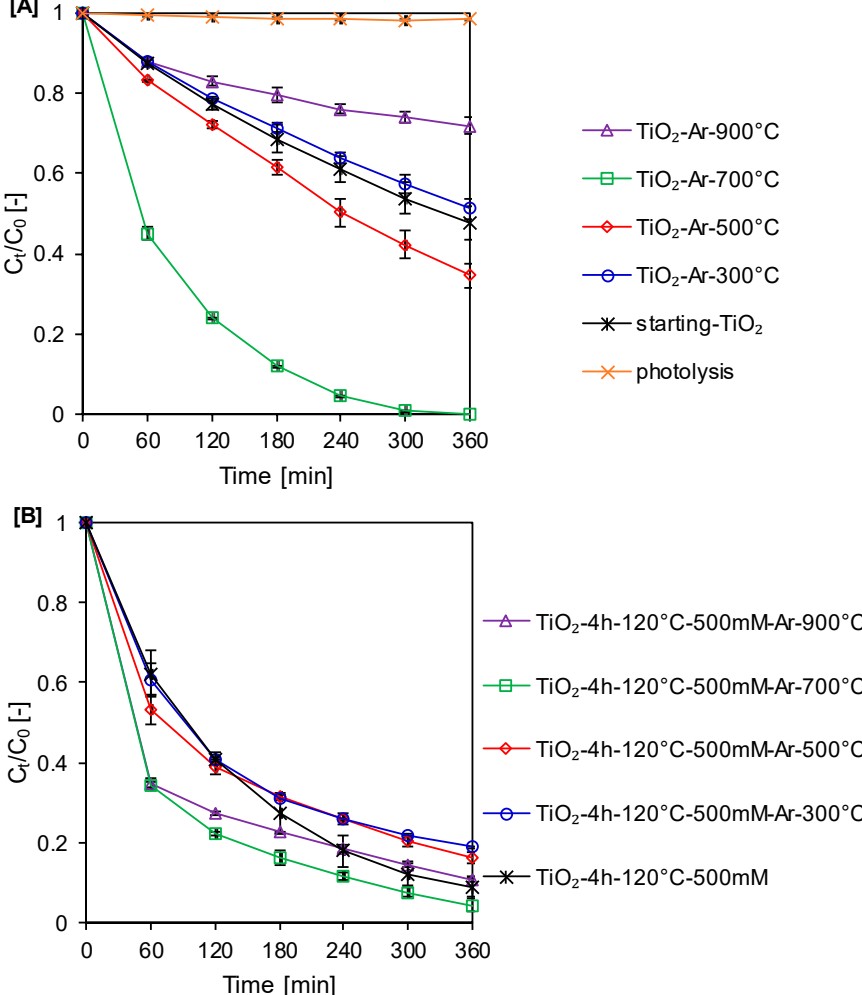

**Figure 7.** Methylene blue decomposition under UV irradiation for starting $TiO_2$ and calcined reference samples (**A**), and APTES-modified $TiO_2$ before and after calcination (**B**).

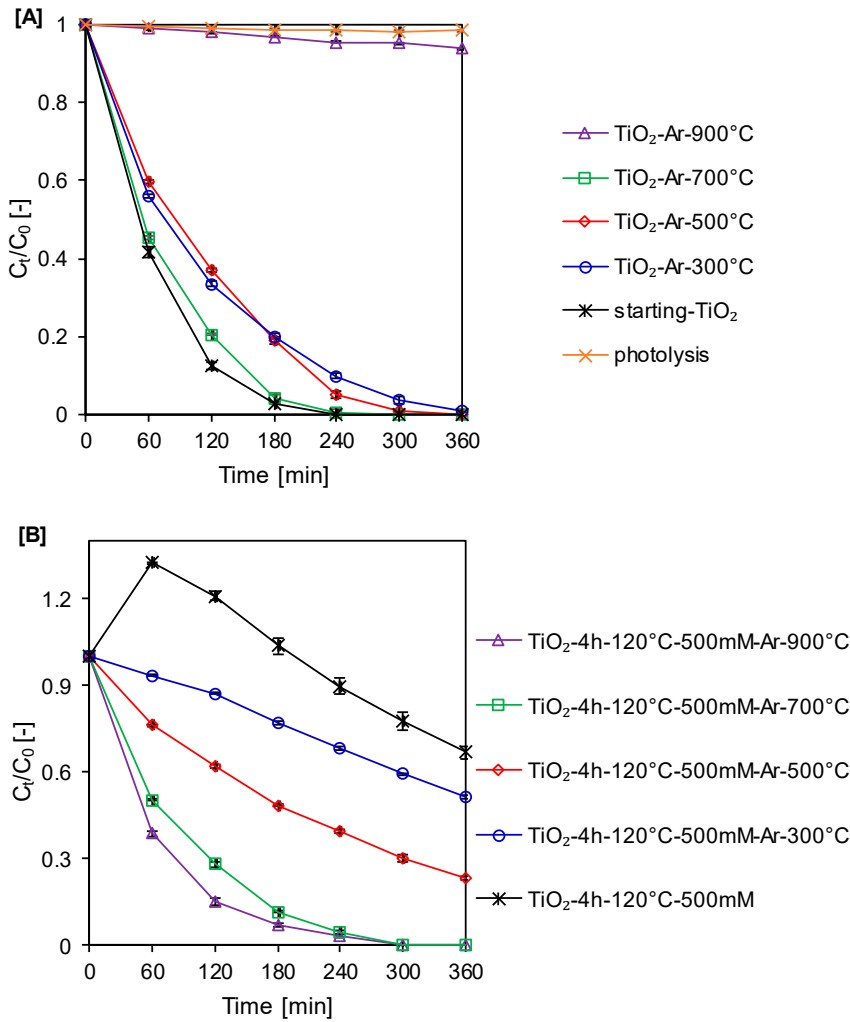

**Figure 8.** Orange II decomposition under UV irradiation for starting TiO$_2$ and calcined reference samples (**A**), and APTES-modified TiO$_2$ before and after calcination (**B**).

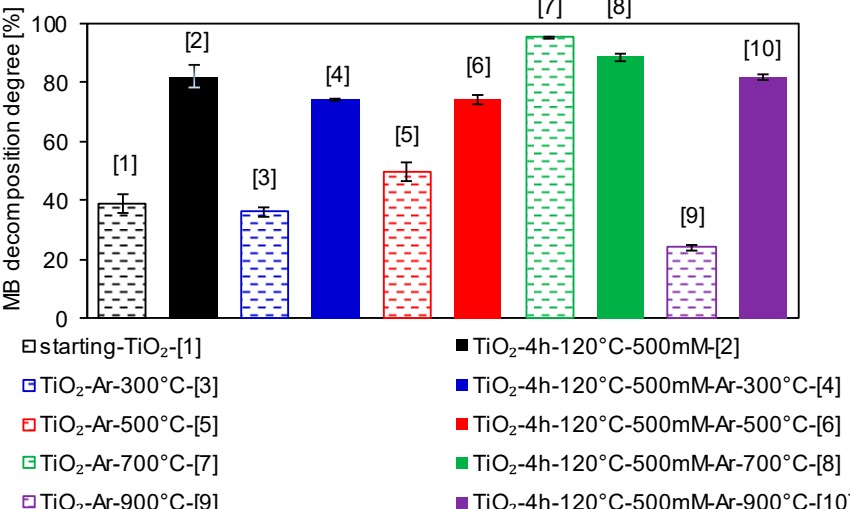

**Figure 9.** Methylene blue decomposition degree after 240 min of UV light irradiation for starting TiO$_2$, calcined reference samples, and APTES-modified TiO$_2$ prior to and after process.

Furthermore, the high porosity facilitated the mass transfer of reagents such as reaction by-products or oxygen, contributing to enhancing the photoactivity of the tested calcined

APTES-functionalized TiO$_2$ photocatalysts [75,76]. Moreover, it was found that APTES-modified TiO$_2$ samples calcined at 500–900 °C were characterized not only by a high dye decomposition degree (see Figure 9), but also by excellent adsorption properties (see Figure 5). These observations were consistent with the reports indicating that the adsorption process strongly influences the photocatalytic degradation reaction [20,24,77]. Furthermore, it was noted that APTES/TiO$_2$ nanomaterial calcined at 900 °C showed the highest efficiency enhancement concerning the reference material. Methylene blue decomposition degree of TiO$_2$-4h–120°C–500mM-Ar–900°C increased by 58% compared to TiO$_2$-Ar-900°C. Since both photocatalysts showed similar zeta potential values (see Table 2), the noticed gain of activity was mostly related to the suppression of anatase-to-rutile phase transformation, thus the higher value of the S$_{BET}$ of the modified samples (see Table 1).

In the case of Orange II, after calcination at 300 °C and 500 °C, a slight decrease (approximately 10% and 5%, respectively) in the photocatalytic activity was noted in comparison to starting TiO$_2$. Significant reduction of the decomposition degree (approximately 95%) was observed for TiO$_2$-Ar–900°C, which was associated with the complete formation of the rutile phase and an increase in the aggregation of molecules, which reduced the specific surface area [30]. After TiO$_2$ modification with APTES, the photocatalytic activity of TiO$_2$-4h–120°C-500mM decreased by 90% after 240 min of irradiation compared to the starting TiO$_2$. For TiO$_2$–4h–120°C–500mM sample, after 60 min of UV light irradiation, significant desorption of dye molecules from the surface of the photocatalyst was noted (see Figure 8B), which may be related to the increase of the temperature of the reaction suspension, caused by the heat generated by the 6 lamps used in the experiments. The significant amount of the Orange II molecules released effectively disrupted the photocatalytic decomposition, causing a decrease in the activity. It was noted that APTES-modified TiO$_2$ calcined at 300 °C and 500 °C exhibited a significantly lower decomposition degree of Orange II than the reference materials prepared at the same temperature. The TiO$_2$-Ar–300 °C and TiO$_2$-Ar–500 °C samples (see Figure 6A) exhibited a higher adsorption degree of Orange II than TiO$_2$–4h–120°C–500mM-Ar–300°C and TiO$_2$-4h–120°C–500mM-Ar–500°C materials (see Figure 6B). The reference samples calcined at 300 °C and 500 °C showed almost identical phase composition and were characterized by a smaller specific surface area than the corresponding APTES-modified TiO$_2$ nanomaterials (see Table 1), the obtained results imply that the adsorption process had a strong influence on the Orange II degradation. Our observations were consistent with the reports indicating that the adsorption process has a substantial impact on the photocatalytic oxidation process and may improve the efficiency of the studied materials [20,24,77]. As the modification temperature increased, the difference between the dye decomposition degree achieved by the reference materials and those modified with APTES decreased. For photocatalysts calcined at 300 °C, the difference in yield was 58%, and at 500 °C it equaled 34%, while at 700 °C the difference in efficiency was only 4%. Only for TiO$_2$-4h–120°C–500mM-Ar–900°C did the sample show a 92% increase in photoactivity after 240 min of exposure to UV light compared to the TiO$_2$-Ar–900 °C (see Figure 10). Similar to the methylene blue decomposition, because both materials exhibited almost the same zeta potential values, the noted increase in the photoactivity was connected with the effective suppression of anatase-to-rutile phase transformation (see Table 1) and still the relatively high specific surface area of TiO$_2$–4h–120°C–500mM-Ar–900°C sample (50 m$^2$/g) in comparison to TiO$_2$-Ar–900°C (3 m$^2$/g) [30,75].

The apparent reaction rate constants were established to better understand the photocatalytic decomposition process of both methylene blue and Orange II. The zero- order, pseudo-first order, and pseudo-second order linear transformations are presented for methylene blue in Figure S2A,B, and for Orange II in Figure S3A–C, respectively. The reaction rate constants were determined after 240 min of exposure to UV light, due to the fact that when the irradiation time was extended to 360 min, after 240 min the points in the graph start to deviate from the typical linear course of the curve (see Figures S2B and S3C). The reported reduction in the reaction rate was related to the formation of intermediates during the decomposition of dyes and the adsorption of these carbon deposits on the

surface of the semiconductor. To avoid impairing the visibility of all graphs, the kinetics of the decomposition after 360 min of irradiation were presented only for the pseudo-second order reactions.

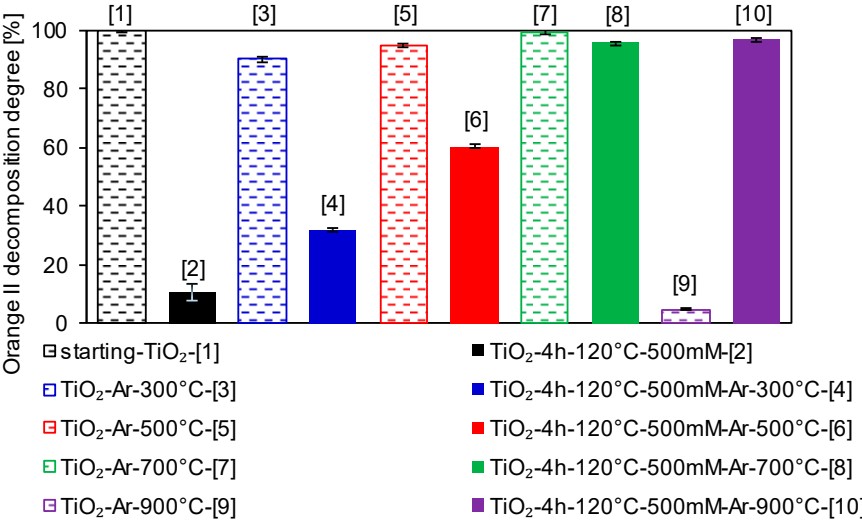

**Figure 10.** Orange II decomposition degree after 240 min of UV light irradiation for starting $TiO_2$, calcined reference samples, and APTES-modified $TiO_2$ before and after calcination.

It was found that the degradation of methylene blue in the case of starting-$TiO_2$, the reference materials calcined at 300–700 °C [30] and $TiO_2$–4h–120°C–500mM followed the Langmuir–Hinshelwood pseudo-first order model. According to the pseudo-second- order model for $TiO_2$-Ar–900°C and all APTES/$TiO_2$ nanomaterials after calcination proceeds. It was noted that after $TiO_2$ functionalization via APTES combined with thermal modification, the kinetics of the methylene blue decomposition changed from pseudo-first to the pseudo-second order. The calculated reaction rate constants were listed in Table S3.

The degradation of Orange II in the case of $TiO_2$–4h–120°C–500mM, $TiO_2$-Ar-500°C and $TiO_2$-4h–120°C–500mM-Ar–300°C followed the zero-order model, while for $TiO_2$-Ar–900°C proceeds according to the pseudo-second-order model. For all other photocatalysts, the decomposition was in accordance with the pseudo-first order model. Based on the data shown in Table S4, the obtained values of $k_0$ were 0.0171 mg/(L·min) for $TiO_2$-4h–120°C–500mM sample, 0.0337 mg/(L·min) for the reference material and 0.0175 mg/(L·min) for $TiO_2$-modified APTES calcined at 300 °C. The noted values of $k_1$ were between 0.0019 L/min and 0.0143 L/min. For $TiO_2$-Ar–900°C sample, $k_2$ equalled 0.0001 L/(min·mg).

The Reusability Tests

Determining the stability of photocatalysts in subsequent cycles of pollutants decomposition plays a crucial role in assessing the possibility of their industrial use. The performance of the studied nanomaterials was tested for four subsequent photocatalytic cycles. For methylene blue decomposition, the results obtained were shown in Figure 11A–E, and for Orange II in Figure 12A–E. In the case of methylene blue, the modification in the Ar atmosphere at 500 °C ($TiO_2$–4h–120°C–500mM-Ar–500°C) and 700 °C ($TiO_2$-4h–120°C–500mM-Ar–700°C) contributed to an improvement in the stability compared to $TiO_2$–4h–120°C–500mM sample. After the first cycle, the decomposition degree of $TiO_2$-4h–120°C–500mM reduced by about 31%, whereas after calcination the photocatalytic activity decreased by about 21% and 26%, respectively. The increase in $S_{BET}$ and $V_{total}$ after calcination (see Table 1) improved the performance of subsequent dye decomposition cycles. Moreover, DRIFT spectra presented in Figure 3, as well as the results of the C and N content shown in Table 2, confirmed the absence of carbon and nitrogen in the materials obtained after the calcination process, the leaching of which reduced the effectiveness of $TiO_2$-4h–120°C–500mM sample [74].

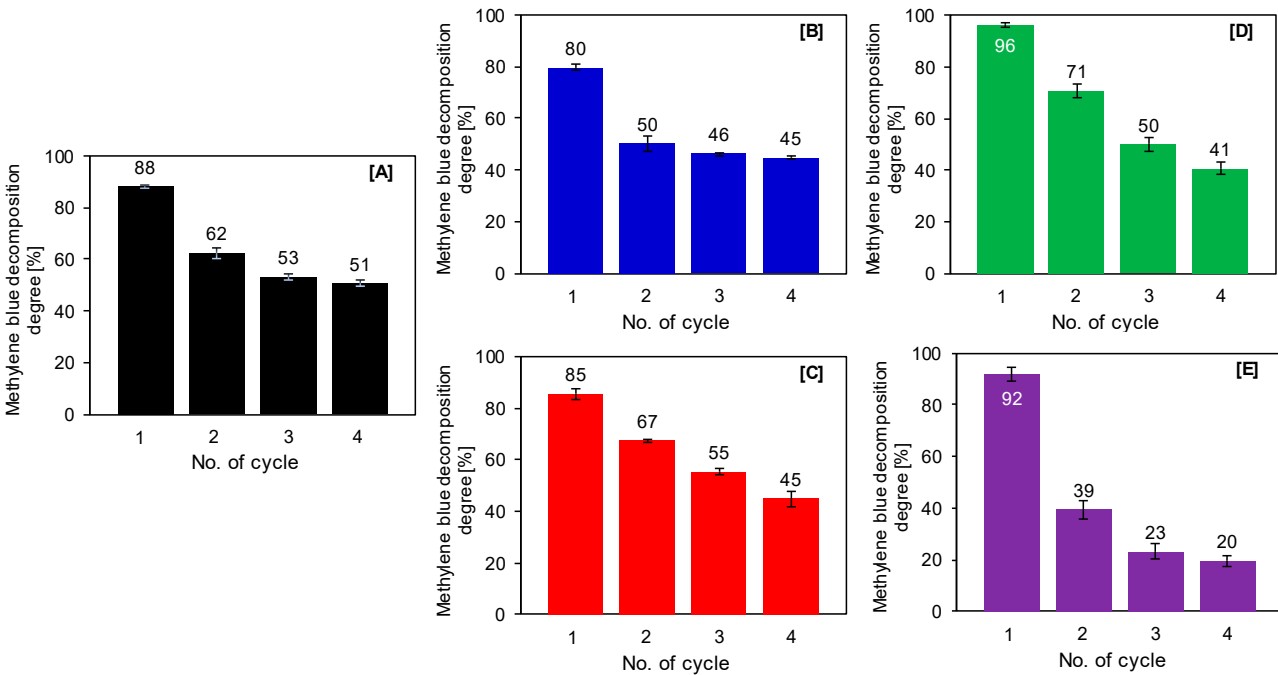

**Figure 11.** Photocatalytic activity in subsequent cycles of methylene blue degradation of $TiO_2$–4h–120°C–500mM (**A**), $TiO_2$–4h–120°C–500mM-Ar-300°C (**B**), $TiO_2$–4h–120°C–500mM-Ar-500°C (**C**), $TiO_2$–4h–120°C–500mM-Ar–700°C (**D**), and $TiO_2$–4h–120°C-500mM-Ar-900°C (**E**) materials.

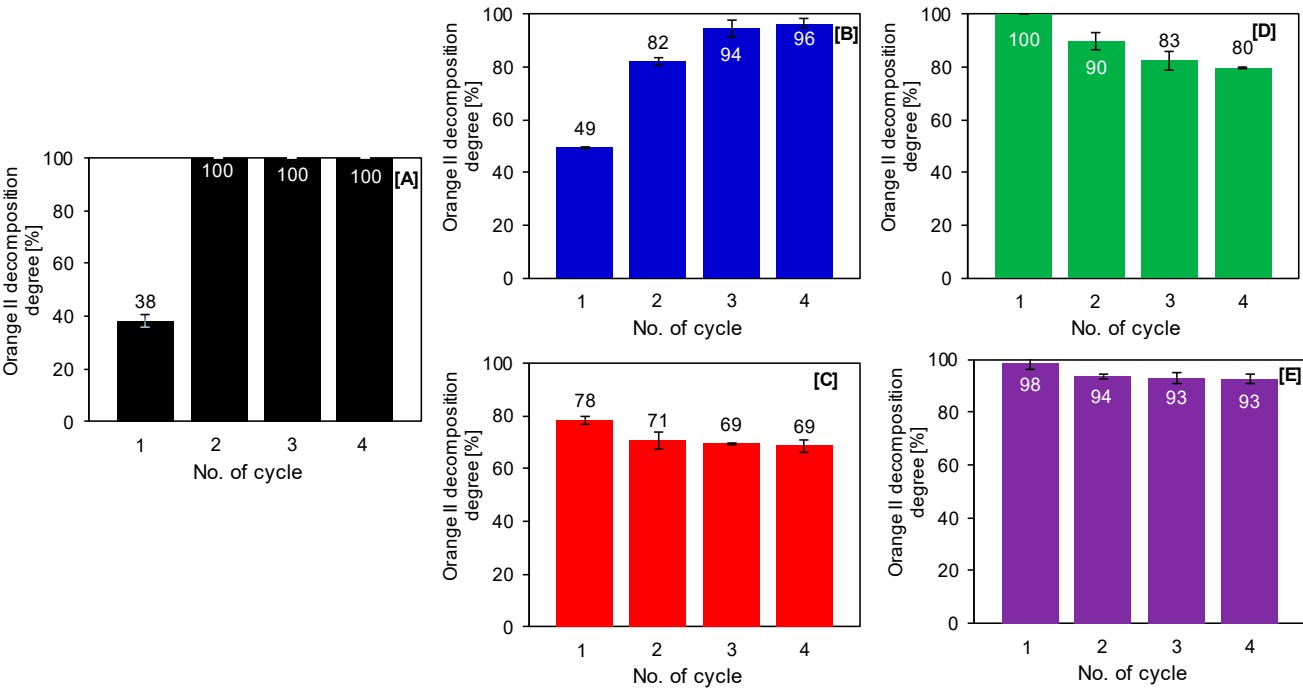

**Figure 12.** Photocatalytic activity in subsequent cycles of Orange II degradation of $TiO_2$-4h–120°C–500mM (**A**), $TiO_2$–4h–120°C–500mM-Ar-300°C (**B**), $TiO_2$–4h-120°C-500mM-Ar-500°C (**C**), $TiO_2$–4h–120°C–500mM-Ar–700°C (**D**), and $TiO_2$-4h–120°C–500mM-Ar-900°C (**E**) nanomaterials.

Additionally, to explain the decrease in the efficiency of calcined materials, after the last fourth cycle of methylene blue decomposition, DRIFT spectra of selected photocatalysts were measured and shown in Figure 13A in combination with the spectra of samples before the exposure process. The observed change in the shape of the spectra after four cycles indicates that the structure of the semiconductor has changed during the photocatalytic

reaction. An apparent increase in the intensity of the band located from 1200 cm$^{-1}$ to 1600 cm$^{-1}$ suggested that in subsequent cycles carbon deposits originating from methylene blue appeared, which caused a decrease in the photocatalytic activity [78].

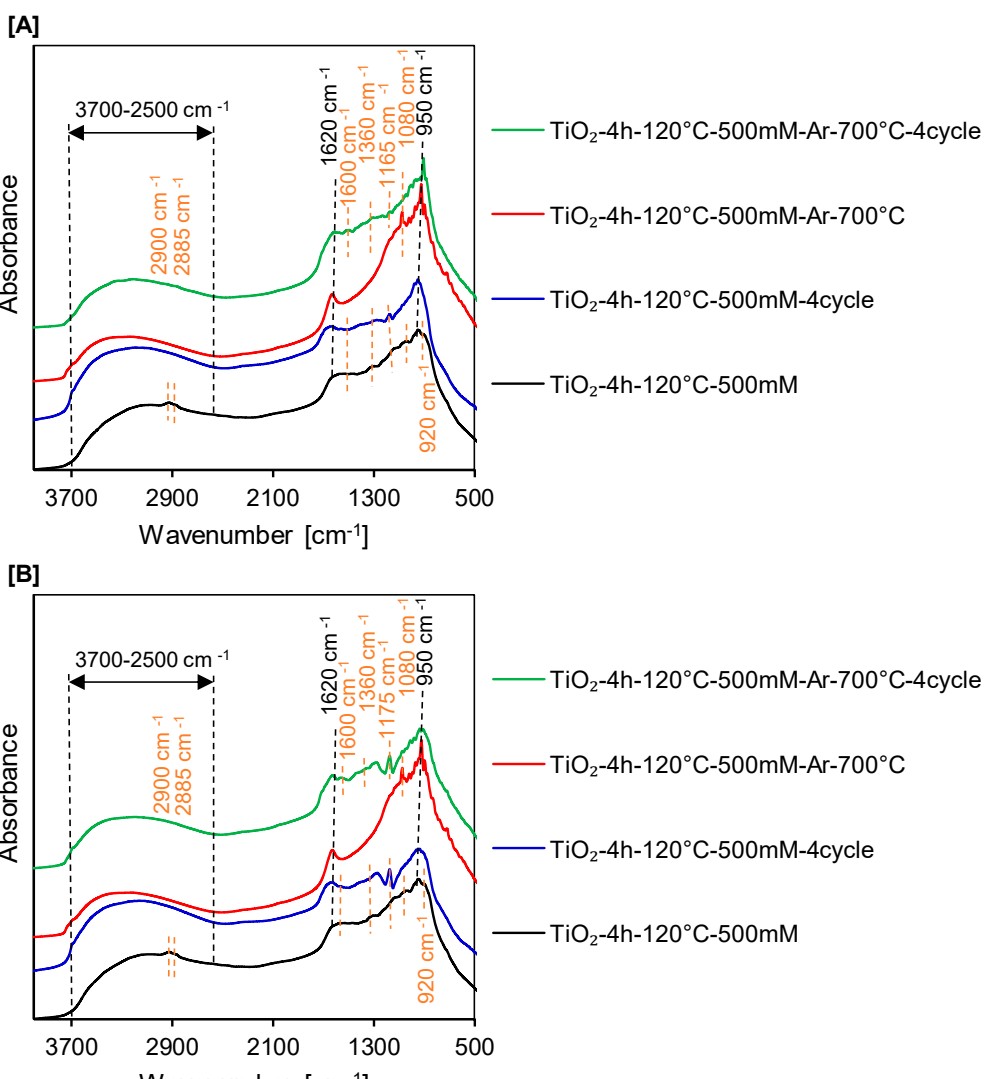

**Figure 13.** DRIFT spectra of TiO$_2$–4h–120°C–50 mM and TiO$_2$–4h–120°C–500mM-Ar–700°C photocatalysts before the first cycle and after the fourth cycle for methylene blue (**A**), and Orange II (**B**) decomposition.

For TiO$_2$-4h–120°C–500mM and TiO$_2$-4h–120°C–500mM-Ar–300°C samples, a significant enhancement in yield was noted in the subsequent cycles of Orange II decomposition (see Figure 12A,B). Compared to the first cycle, the decomposition degree of TiO$_2$–4h–120°C–500mM increased from 38% to 100% after the second cycle and remained at this level even after the last cycle. For TiO$_2$–4h–120°C–500mM-Ar–300°C there was a gradual increase in activity from 49% after the first cycle, through 82% after the second to 96% after the fourth cycle. The materials calcined at 500 °C and 900 °C had relatively high stability, because after four cycles only a slight decrease in the activity of 9% and 5%, respectively, was recorded. Only a TiO$_2$–4h–120°C–500mM-Ar–700°C sample exhibited a slight reduction in the activity during the reusability test, for which the decomposition degree decreased from 100% after the first cycle to 80% after the fourth cycle. The increase in the activity and high stability of photocatalysts in the subsequent cycles of Orange II decomposition resulted from the appearance of −SiO$_3$$^-$ groups on the TiO$_2$ surface, which were observed on DRIFT spectra at 1175 cm$^{-1}$ (see Figure 13B) [78]. According to the

literature [17], the presence of the $-SO_3^-$ group could be assumed to improve the removal efficiency. An increase in the intensity of the band located from 1200 $cm^{-1}$ to 1600 $cm^{-1}$ indicated that in subsequent cycles semiproducts originating from Orange II decomposition appeared, which contributed to a slight decrease in the efficiency of the sample calcined at 700 °C. Apart from the confirmed significant influence of the phase composition on the photocatalytic efficiency, the obtained FT-IR/DR spectra demonstrated that the reduction of the dye decomposition degree was strongly influenced by the adsorption of methylene blue on the surface of the tested nanomaterials. A negatively charged surface of APTES-modified photocatalysts impacts the sorption of degradation co-products of methylene blue in each next cycle, thus decreasing the photoactivity. On the other hand, the presence of the negative-loaded groups on the surface of tested materials (i.e. $SiO_3^-$ groups) inhibits sorption of Orange II degradation semiproducts. Therefore, the adsorption process also plays an essential role in the photocatalytic decomposition of pollutants [17,79,80].

## 3. Materials and Methods

### 3.1. Materials and Reagents

Crude $TiO_2$ pulp, purchased from the chemical plant Grupa Azoty Zakłady Chemiczne "Police" S.A. (Police, Poland), was used as a $TiO_2$ source. Before modification, the raw pulp was pre-prepared to reach pH equal to 6.8 due to the presence of post-production sulphuric acid remains. This stage was described in detail in our previous article [61]. The received material was named as starting $TiO_2$. 3-aminopropyltriethoxysilane (APTES, $C_9H_{23}NO_3Si$, ≥98%) from Merck KGaA (Darmstadt, Germany) was used as a modifier of the $TiO_2$. Ethyl alcohol (purity 96%, pure p.a.) purchased from P.P.H. "STANLAB" Sp.J. (Lublin, Poland) was used as a solvent of APTES. During photocatalytic decomposition tests, Orange II ($C_{16}H_{11}N_2NaO_4S$, ≥85%, Firma Chempur®, Piekary Śląskie, Poland) and methylene blue ($C_{16}H_{18}ClN_3S$, ≥82%, Firma Chempur®, Piekary Śląskie, Poland) were used as model organic water pollution.

### 3.2. Synthesis of 3-Aminopropyltriethoxysilane (APTES)-Modified $TiO_2$

The APTES/$TiO_2$ nanomaterials were obtained by the solvothermal process and calcination. In the beginning, 5 g of starting $TiO_2$ was dispersed in 25 mL of APTES solution. The concentration of the modifier in ethanol was 500 mM. Next, the prepared mixture was modified in a pressure autoclave for 4 h at 120 °C, ensuring continuous stirring at 500 rpm. Then, the suspension obtained was rinsed with ethanol and distilled water to eliminate all residual chemicals. The sample was dried at 105 °C for 24 h in a lab dryer. The material obtained was denoted as $TiO_2$–4h–120°C–500mM. Finally, the prepared material was heated in argon atmosphere (purity 5.0, Messer Polska Sp. z o.o., Chorzów, Poland). The quartz crucible containing the obtained photocatalyst was placed inside a quartz tube in the middle of the GHC 12/900 horizontal furnace (Carbolite Gero, Ltd., Sheffield, UK). Calcination was conducted in the range of 300–900 °C ($\Delta t$ = 200 °C). Prior to the heating step, the argon was passed through the quartz tube for 30 min to remove the air present in the quartz pipe. After that, the furnace was heated up to the set temperature in the argon flow of 180 mL/min. The calcination time was 4 h. After the set time, the furnace was slowly cooled down to room temperature. Nanomaterials received after the thermal modification of starting $TiO_2$ in the Ar atmosphere were named as reference samples denoted as $TiO_2$-Ar-t, while APTES/$TiO_2$ materials obtained after heat treatment were marked as $TiO_2$–4h–120°C–500mM-Ar-t, where *t* is the calcination temperature.

### 3.3. Characterization Methods

The low-temperature nitrogen adsorption-desorption measurements (conducted at 77 K) carried out on the QUADRASORB evo™ Gas Sorption analyzer (Anton Paar GmbH, Graz, Austria) were used to calculate the Brunauer–Emmett–Teller (BET) specific surface area and pore volume. Before the measurements, all samples were degassed for 12 h at 100 °C under high vacuum to remove any residual contaminants present on the sur-

face of tested materials. The single-point value determined the total pore volume from the nitrogen adsorption isotherms at relative pressure $p/p_0$ = 0.99. Micropore volume was estimated using the Dubinin–Radushkevich method, while mesopore volume was determined as the difference between $V_{total}$ and $V_{micro}$. The zeta potential values were determined with the ZetaSizer NanoSeries ZS (Malvern PANalytical Ltd., Malvern, UK). The crystalline structure of the tested photocatalysts was identified by the X-ray powder diffraction analysis (Malvern PANalytical B.V., Almelo, Netherlands) using Cu Kα radiation ($\lambda$ = 1.54056 Å). Scherrer's equation was used to calculate the mean crystallites size. To identify the phase composition, the PDF-4+ 2014 International Centre for Diffraction Data database (for anatase: 04-002-8296 PDF4+ card, and for rutile: 04-005-5923 PDF4+ card) was applied. The FT-IR-4200 spectrometer (number of scans 100, resolution 4.0 cm$^{-1}$, JASCO International Co. Ltd., Tokyo, Japan) equipped with DiffuseIR accessory (PIKE Technologies, Fitchburg, WI, USA) was utilized to determine the surface functional groups and to notice the surface changes during the reusability test. The V-650 UV-Vis spectrophotometer (JASCO International Co., Tokyo, Japan) equipped with a PIV-756 integrating sphere accessory for measuring DR spectra (JASCO International Co., Tokyo, Japan) was used to examine the light reflectance abilities of the new APTES/TiO$_2$ samples. Spectralon$^{®}$ Diffuse Reflectance Material (Labsphere, New Hampshire, NE, USA) was applied as the reference material. The optical bandgap energy ($E_g$) of the samples was calculated by plotting $[F(R)h\upsilon]^{1/2}$ as a function of photon energy and next extrapolating the linear parts to $[F(R)h\upsilon]^{1/2}$ = 0 [81]. The CN 628 elemental analyzer (LECO Corporation, St. Joseph, MI, USA) was used to determine the total carbon and nitrogen content in the studied nanomaterials. The certified ethylenediaminetetraacetic acid (EDTA) standard (LECO Corporation, St. Joseph, MI, USA) containing 9.56 ± 0.03 wt.% of nitrogen and 41.06 ± 0.09 wt.% of carbon was utilized to prepare the calibration curve for preparing calibration curves. The error range for measurements was maximally ±0.1%.

### 3.4. Adsorption Capacity

To determine the adsorption capacity, the adsorption experiments were carried out in Erlenmeyer flask with starting-TiO$_2$, reference samples and APTES-modified TiO$_2$ nanomaterials, by stirring at 150 rpm 0.125 g of the photocatalyst in 0.25 L of methylene blue and Orange II aqueous solutions with concentrations of 1, 2, 4, 6, 8, 10, 12 and 15 mg/L, under light-free conditions in a thermostatic chamber at 20 °C (Pol-Eko-Aparatura sp.j., Włodzisław Śląski, Poland). This process continued for 240 min to establish the adsorption-desorption equilibrium between the dye and photocatalysts surface. Ultrapure water (Merck Millipore Sp. z o.o., Warszawa, Poland) was used to prepare all solutions. After 240 min, 10 mL of the withdrawn suspension was centrifuged to separate all suspended nanoparticles. The concentration of methylene blue and Orange II was analyzed by the V-630 UV-Vis spectrometer (JASCO International Co., Tokyo, Japan). The experimental equilibrium adsorption data were analyzed using Freundlich [82], Langmuir [72], Langmuir–Freundlich [70], and Temkin [71] isotherm models. The Statistica software (version 13.1) was used to plot the adsorption isotherms.

### 3.5. Photocatalytic Activity Test

Methylene blue and Orange II were selected for the photocatalytic activity and reusability tests. Photocatalytic decomposition was carried out under UV-Vis light with the radiation intensity of 65 W/m$^2$ for 300–2800 nm and 36 W/m$^2$ for the 280–400 nm region, supplied by a series of 6 lamps of 20 W each (Philips, Amsterdam, Netherlands). Because of the low intensity of visible (Vis) irradiation used, this type of light was named as UV light. The experiments were performed in a 0.6 L glass beaker using 0.5 L of dye solution, with an initial concentration of 15 mg/L and the concentration of photocatalyst equalled 0.5 g/L. The experimental procedure consisted of two steps. Firstly, before the irradiation stage, the prepared suspension was magnetically stirred under dark conditions to establish the adsorption-desorption equilibrium between the TiO$_2$ surface and dye molecules. The

time necessary to achieve the adsorption-desorption equilibrium was determined based on research on the sorption properties of the photocatalysts. Secondly, the suspension was exposed to UV light irradiation. The total exposure time was 360 min. The dye concentration was measured every 60 min with the V-630 UV-Vis spectrometer (JASCO International Co., Tokyo, Japan). Prior to each measurement, 10 mL of the withdrawn suspension was centrifuged to remove all suspended photocatalyst nanoparticles. Additionally, the reusability test was determined based on the decomposition of dyes under UV light after 360 min of irradiation. After each cycle, the material was separated by filtration and dried for 12 h at 105 °C and then added to a new dose of the dye solution with appropriate initial concentration. Based on the results obtained, the reaction rates of model organic pollutants decomposition were determined by adjusting the appropriate order of the reaction. The zero reaction rate constant was established according to the zero-order model [47].

The pseudo-first reaction rate constant was calculated using the Langmuir– Hinshelwood pseudo-first order model and the pseudo-second reaction rate constant was determined following the pseudo-second order model [83].

## 4. Conclusions

The APTES/TiO$_2$ nanomaterials were obtained by a solvothermal process at 120 °C, and subsequent calcination in Ar atmosphere in the range of 300 °C–900 °C. In the present article, for the first time, the role of adsorption in the photocatalytic decomposition of dyes on APTES-modified TiO$_2$ nanomaterials, as well as the influence of calcination in the inert gas atmosphere on the adsorption capacity and stability of APTES/TiO$_2$ samples were determined. The presence of the modifier on the TiO$_2$ surface was confirmed by DRIFTS measurements and carbon and nitrogen content analysis. It was found that the Langmuir and Langmuir–Freundlich isotherm models, due to their highest fit, best describe the adsorption of methylene blue on the surface of the examined APTES-modified photocatalysts, while for the Orange II adsorption process, the best fit was found for Freundlich and Langmuir–Freundlich isotherm models. The recorded alterations in the adsorption capacity were related to the changes in the surface charge and the S$_{BET}$ of the APTES-modified TiO$_2$. It was also observed that the adsorption process had a significant impact on the photooxidation of dyes. The APTES/TiO$_2$ photocatalysts calcined at 900 °C, showed a markedly higher methylene blue and Orange II degradation degree in comparison to calcined reference materials, which was related to the fact that functionalization via APTES, which is a good source of silicon, effectively suppressed anatase-to-rutile phase transition, as well as the growth of crystallites size. Furthermore, in the case of methylene blue, thermal modification at 500 °C and 700 °C contributed to an improvement in the stability compared to TiO$_2$–4h–120°C–500mM sample for Orange II, all photocatalysts showed high efficiency during the reusability test.

**Supplementary Materials:** The following are available online at https://www.mdpi.com/2073-4344/11/2/172/s1, Figure S1: Pore size distribution plots of starting-TiO2, calcined reference samples (A), and APTES-modified TiO2 photocatalysts (B), Figure S2: The pseudo-first-order plot (A), and the pseudo-second-order plot (B) of methylene blue decomposition., Figure S3: The zero-order plot (A), the pseudo-first-order plot (B), and the pseudo-second-order plot (C) of Orange II decomposition. Table S1: Isotherm constants for the adsorption process of methylene blue on starting TiO2, calcined reference samples and APTES-modified photocatalysts. Table S2: Isotherm constants for the adsorption process of Orange II on starting TiO2, calcined reference samples and APTES-modified photocatalysts. Table S3: The fitting parameters, the pseudo-first and pseudo-second reaction rate constants for methylene blue photoremoval (after 240 min of UV radiation). Table S4: The fitting parameters, zero, pseudo-first, and pseudo-second reaction rate constants for Orange II photoremoval (after 240 min of UV radiation).

**Author Contributions:** Conceptualization, A.S., A.W. and A.W.M.; Formal analysis, E.K.-N.; Funding acquisition, A.W.M.; Investigation, A.S., A.W. and P.R.-K.; Methodology, A.S. and A.W.; Software, A.S. and E.K.-N.; Supervision, E.K.-N., A.W. and A.W.M.; Writing—original draft, A.S. and E.K.-N.; Writing—review and editing, E.K.-N., A.W., A.S., A.W.M. All authors have read and agreed to the published version of the manuscript.

**Funding:** This research was funded by the National Science Centre, Poland, grant no. 2017/27/B/ST8/02007.

**Conflicts of Interest:** On behalf of all authors, the corresponding author states no conflict of interest.

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
