# Peer review of "The Role of Adsorption in the Photocatalytic Decomposition of Dyes on APTES-Modified TiO2 Nanomaterials"

_catalysts, doi:10.3390/catal11020172_

Round 1

Reviewer 1 Report

Review of the Manuscript No Catalysts-1058580

This work is dedicated to synthesis of novel aminosilane-modifeid titania nanomaterials and their successful application for photocatalytic decomposition of methylene blue and Orange II dyes.

The samples were characterized by modern techniques like BET, XRD, FTR/DRS, etc. The presence of aminosilane on the TiO2 surface was confirmed by FT-IR/DRS measurements and elemental analysis. It was found that both the Langmuir-Freundlich and Temkin isotherm models, best fit and describe the adsorption of methylene blue and Orange II on the surface of the examined photocatalysts.

I recommend acceptance of the paper after minor revision.

Meanwhile I have some questions, notes and recommendations.

  1. Please correct English text of the Abstract, Introduction, Results, and Conclusions.

Some sentences in the Conclusions are the same as in the Abstract. This should be corrected.

  1. “Before the measurements, all samples were degassed for 12 h at 100°C under high vacuum to remove any residual contaminants present on the surface of tested materials” / 3. Materials and methods, 3.2. Characterization methods, page 4, Rows 531-533/

Why authors have not used higher temperatures like 200 or 300°C, which would clear the surface more effectively before BET measurements?

  1. The authors should add pore size distribution plots.

  1. Conclusions, page 6, rows 625-627

“The APTES/TiO2 nanomaterials were obtained by solvothermal process at 120°C with the

concentration of APTES equalled 500 mM and calcination in the Ar atmosphere in the range of 300°C-900°C.”

“should be corrected as follows The APTES/TiO2 nanomaterials were obtained by solvothermal process at 120°C, and subsequent calcination in Ar atmosphere in the range of 300°C-900°C.”

  1. “It was found that the Langmuir-Freundlich and Temkin isotherm models, due to their highest fit, best describe the adsorption of methylene blue on the surface of the examined photocatalysts, while for the Orange II adsorption process, the best fit was found for Freundlich and Langmuir-Freundlich isotherm models.” / Abstract, page 1, 17-19 and Conclusions, page 6, rows 631-634/

Should be replaced with:

“It was found that both the Langmuir-Freundlich and Temkin isotherm models, best fit and describe the adsorption of methylene blue and Orange II on the surface of the examined photocatalysts.”

  1. Authors should add more references about different methods for removal of dyes.

Author Response

Dear Reviewer,

Thank you very much for the perceptive review, constructive comments on our manuscript and the time and energy to help us improve the paper. According to your comments, we have rearranged and improved the text of the manuscript with particular emphasis identified in the review excerpts. All changes we made are marked in red in the revised version of the manuscript. We hope that the manuscript so amended in line with the suggestions made by the Referee will prove to be satisfactory and will meet the criteria required for its publication in Catalysts.

Reviewer 2 Report

In the manuscript entitled: " The role of adsorption in the photocatalytic decomposition of dyes on APTES-modified TiO2 nanomaterials" the authors report a novel preparation method for APTES-modified TiOs nanoparticles and the study of the influence of calcination on the adsorption capacity, photocatalytic activity and stability for Methyele blue and Orange II decomposition. Despite the topic is interesting, there are many issues that remain unclear or need further clarification.

For instance, the authors claim in the introduction that APTES-modified TiO2 has been already reported for the photocatalytic degradation of different dyes. Therefore, the only novelty remains in the preparation procedure, however, no comparison between APTES-modified TiO2 from previous methods and the here reported has been supplied, and therefore, the reader cannot know which is the advantage is this novel method, if there is.

The materials characterization is well addressed, however, in the adsorption experiments section, the author claim that for sample TiO2-4h-120-500mM-Ar-900 the adsorption degree reached 30% while  for TiO2-Ar-900 only 5%, and the authors attribute this difference to the suppresion of the phase transformation of anatase to rutile. but these materials also show a larga surface area difference from 50 to 3 m2/g. Despite the phase transition could have some influence, the 10 times larger surface area is probably the main reason of this adsorption difference. This conclusion has been repeated several times in the manuscript without take into consideration the difference in surface area of these 2 materials. Fear comparison must be carried out, and the adsorption should be normalized per surface area.

On the other hand, in the photocatalytic activity test section, the conclusions seems to do not match the experimental evidences. For methylene blue decomposition, the authors must explain why samples at 700 C do not follow the same trend of the other samples. In addition the most efficient sample corresponds to the no modified one at 700C. Moreover, for Orange II not only the most efficient sample is the starting TiO", but also almost all unmodified samples are more efficient that the APTES-modified ones. Therefore, what is the sense to make this modification if bare TiO2 is photocatalytically more active??

Finally, the stability tests show very different behaviour between the differetn dyes. The authors should explain the origin of these differences.

Additionally, the number of Figures should be reduced. Main plots must appear in the main text, but some others should appear as supplementary information.

In overall, the main strength of this manuscritps should be the enhanced photocatalytic activity for APTES-modified TiO2, but it seems that this is not the case for some dyes. The advantages of this preparation method over other methods in the literature should be present too. For that reasons I cannot recommend for publication.

Author Response

Dear Reviewer,

I have tried to correct the manuscript according to your suggestions. Our replies to your comments could be find as attach file. 

Reviewer 3 Report

Kusiak-Nejman et al. presented their finding on “The role of adsorption in the photocatalytic decomposition of dyes on APTES-modified TiO2 nanomaterials”. However, this presented research is too similar with their previous published articles Effect of calcination on the photocatalytic activity and stability of TiO2photocatalysts modified with APTES, Journal of Environmental Chemical Engineering 9 (2021) 104794

https://doi.org/10.1016/j.jece.2020.104794”.

Unfortunately, the presented research is too far given new information and perspective compared to their previous publication and it has too much similarity between each other. I highly recommend the rejection of the submitted manuscript due to these similarity with their own work.

These are some similarity, which I recognized during my peer review process, between submitted manuscript and accepted paper from another journal.

  1. Only the concentration difference is used to generate new data set without given any explanation.

Submitted manuscript:

The APTES/TiO2 nanomaterials were obtained by the solvothermal process and calcination. In the beginning, 5g of starting-TiO2 was dispersed in 25 mL of APTES solution. The concentration of modifier in ethanol was 500 mM. Next, the prepared mixture was modified in a pressure autoclave for 4 h at 120°C, ensuring continuous stirring at 500 rpm…

Accepted paper:

In these studies, the new nanomaterials were obtained from TiO2 and APTES by the solvothermal process and calcination. Firstly, 5 g of starting TiO2 powder was dispersed in 25 mL of APTES solution, the concentration of APTES in ethanol was 2000 mM and then modified in a pressure autoclave for 4 h at 180 ◦C ensuring continuous stirring at 500 rpm….

  1. They used similar data without given any reference!!!

Example 1

Example 2

Example 3

Example 4

  1. Author “ Antoni W. Morawski “ has 11 self-citations in submitted manuscript. But, they didn’t mention about similarity with their previous publication.

Author Response

Dear Reviewer,

despite the negative recommendation for our article, we are still grateful for the time you spent reading this paper. In the attachment I am sending explanations about similarities between the current article and the article we published previously. We, in turn, tried to show the differences between these articles. In the article Effect of calcination on the photocatalytic activity and stability of TiO2 photocatalysts modified with APTES" we did not discussed in detail the role of the adsorption that strongly influenced the effectiveness of the pollutant photodegradation. Taking into account the fact that these aspects have not been considered in the literature so far, we decided to prepare a new group of samples (differing, among others, in autoclave pre-treatment and modifier concentration), for which we performed sorption tests and photocatalytic decomposition of selected pollutants. We did not use exactly the same group of samples that we described earlier in the published article. It was more important for us to present results showing adsorption properties and their influence on the efficiency of the photocatalytic decomposition of colour contaminants, which we did not discuss in the previous article.

The Figures presenting data for starting TiO2 and calcined reference samples, and showed in our already publised article, were removed from the manuscript. We admit that was a huge mistake that should not have happened. We can only honorably admit it, apologize and remove it from the article.

Round 2

Reviewer 2 Report

The authors have now reviewed and edited the manuscript, improving its quality. The main aim of the manuscritp is now more clear. Despite this, the manuscript is still very long and difficult to follow. Tables 3-6 should be included in supplementary information, and the extention of the text must be summarized. Moreover, convenient comparison with other results in the literature must be added. For instance, it has been demonstrated that the decrease in adsorption has a detrimental effect on the photocatalytic activity. Therefore, comparison with similar materials , but prepared by means of other methods, must be carried out in order to confirm this observations.

Author Response

Dear Reviewer,

according to your suggestions we moved Tables 3-6 to Supplementary Information. Secondly, we add the santence you wrote. However, it is difficult to compare our results with the literature data due to the fact that our work is the first paper discussing the impact of adsorption on photoctalytic oxidation process. Adsorption properties of rutile TiO2 functionalized inter alia with APTES molecules was tested by Andrzejewska et al. (Andrzejewska, A.; Krysztafkiewicz, A.; Jesionowski, T. Adsorption of organic dyes on the aminosilane modified TiO2 Dyes Pigm. 2004, 62, 121–130). Due to the limited photocatalytic activity of rutile pigments, no photocatalytic activity was examined. We add short explamation in the text (lines 69-71). 

Additionally, many revisions were clearly highlighted, using the "Track Changes" function in Microsoft Word, so that they should be easily visible to you. 

Yours faithfully, 

Ewelina Kusiak-Nejman